# Evaluation of Carbon Nanotube Incorporation in Cementitious Composite Materials

**DOI:** 10.3390/ma12091504

**Published:** 2019-05-08

**Authors:** Ana Catarina Jorge Evangelista, Jorge Fernandes de Morais, Vivian Tam, Mahfooz Soomro, Leandro Torres Di Gregorio, Assed N. Haddad

**Affiliations:** 1Engineering and Mathematics, School of Computing, Engineering and Mathmatics, Western Sydney University, Sydney 2751, Australia; v.tam@westernsydney.edu.au (V.T.); m.soomro@westernsydney.edu.au (M.S.); 2Department of Telecommunications Engineering, Universidade Federal Fluminese, Niterói 24.020-971, Brazil; assed@poli.ufrj.br; 3Program de Engenharia Ambiental, Universidade Federal do Rio de Janeiro, Rio de Janeiro 21941-901, Brazil; leandro.torres@poli.ufrj.br

**Keywords:** cementitious materials, carbon nanotubes, compressive strength, non-destructive test, SEM

## Abstract

Over the last decades, new materials with outstanding performance have been introduced in the construction industry. Considering these new technologies, it is worth mentioning that nanotechnology has revolutionized various areas of engineering. In the area of civil engineering and construction, cement is used for various purposes and the search to improve its performance has been receiving growing interest within the scientific community. The objective of this research was to evaluate the behavior of cement mortar produced by the addition of multi-walled carbon nanotubes (MWCNTs) in different concentrations by comparing their physical and mechanical properties with the properties of the nanotube-free composite. Motivated by the lack of consensus in the literature concerning to the optimal dosage of CNTs in cementitious matrices, three different carbon nanotube ratios, 0.20, 0.40 and 0.60 wt % Portland cement, were investigated with the aim of evaluating the mechanical properties. Destructive tests were carried out to determine the compressive strength, flexural strength and split tensile strength. Additionally, a non-destructive test was performed to determine the dynamic elastic modulus and density. Scanning electron microscopy (SEM) images showed the interaction between the MWCNTs and the hydration products of Portland cement mortar. The results indicated the potential contribution of 0.40 wt % cement CNTs to the enhancement of the mechanical properties of the cement composite as a promising construction material.

## 1. Introduction

Over the past decades, investigations of construction material properties have only been possible on a macro scale. However, new materials have been developed and knowledge of the nano scale behavior is imperative. For example, research of the cement matrix and its interaction with other concrete design constituents is a powerful approach to the development of concretes with better properties and more controlled degradation.

The construction industry is a branch of engineering that is of great importance to society, ranked among the top forty industrial sectors likely to be soon affected by nanotechnology. Concrete and its related products are considered essential inputs for this industry [1,2].

On the other hand, since 1991, carbon nanotubes have been used by several industries (electronic, automobile, and astronautic) in the research of nano-size particles. The carbon nanotubes (CNTs) have outstanding mechanical, thermal and electrical properties including an elastic modulus of 1 TPa, tensile strength of 63 GPa, thermal conductivity of 6600 W m^−1^ K^−1^, electrical resistivity in the order of 10^−4^ Ω-cm, and a current density capacity up to 106 A cm^2^ [3]. Carbon nanotubes can be found in two major forms: single-walled carbon nanotubes and multi-walled carbon nanotubes [1]. Another carbon structure is the nanotube in which carbon atoms are connected in the form of hollow tubes, such as fullerenes, with diameters of one to several tens of nanometers. Carbon atoms can also be combined in nanosheets and nanofites, which have membrane-like structures a few nanometers thick [4].

Understanding, at the nano scale, the performance of the cement matrix and its interaction with other components can be a powerful step in the development of superior concrete with enhanced properties and a more effectively controlled deterioration process. Concrete requires excellent filling capacity and adequate resistance to segregation. In general, conventional research on concrete considers the effects of the addition of different substances, such as ash, ground granulated blast-furnace slag, and limestone, on the performance of the mechanical properties [5].

Experiments with nanomaterials have already allowed for the development of lower cost high-performance cement compounds used extensively in civil engineering [2,6]. The properties of concrete in its initial state, such as its fluidity and workability, are governed by the distribution and size of the particles, while the properties of concrete in its hardened state, such as strength and durability, are affected both by the size of the particles and by the arrangement or grouping of the aggregate [7].

One of the most desirable properties of the nanomaterials used in the construction industry is their ability to confer mechanical reinforcement to the concrete structure. Due to the remarkable properties of carbon nanotubes, this can be achieve by the incorporation of this material into cement compounds, resulting in a new class of cement products valuable to the construction industry [8,9]

The nanotubes modify the molecular structure of the cementitious materials, leading to improved physical and mechanical properties. These nanotubes directly influence the mechanical performance, volumetric stability, durability and sustainability of the concrete structure [10,11]. 

Nowadays, carbon nanotubes are also being used to reinforce many types of materials including metals [12]. Studies have shown that pressure above 500 MPa can be transferred across the interface between a polymer and carbon nanotube, this strength is 10 times higher than that between polymer and carbon fibers [13].

Makar et al. [14] presented hardness measurements, evidence that carbon nanotubes can affect cement hydration in their initial state and that a strong bond is possible between cement paste and carbon nanotubes. They reported that nanotubes have a special role in controlling the cracking of the composite as, in terms of the size and proportion incorporated into the matrix, they are dispersed better than the fibers used as reinforcement. Agullo et al. [15] observed an increase in early age compressive strengths by adding a low concentration of multi-walled carbon nanotubes (MWCNTs) to the cement composites.

Balaguru and Chong [16] believed that the development of nanoscience for concrete was needed since properties such as low shrinkage, resistance to high and low temperatures, compatibility with different types of fibers, and responsiveness to nanomaterials such as carbon nanotubes can be used to create new products with far better performance. Carbon nanotubes and carbon nanofibers appear to be some of the most promising nanomaterials to improve the mechanical properties of cementitious material due to their resistance to crack propagation and other abnormalities [17].

Hunashyal et al. [18] noted higher tensile strength and a better stress–strain relationship, in direct tension, for cement samples containing carbon nanotubes. Han et al. [19] showed that the addition of multi-walled carbon nanotubes (MWCNTs) could decrease the water absorption and permeability coefficient of reinforced cement-based materials. Manzur et al. [20] found that, when used in the mortar, MWCNTs with an OD (outside diameter) of 20 nm or less obtained higher compressive strengths, compared to larger OD MWCNTs, while the maximum compressive strength was achieved by the smallest size MWCNTs considered in the study.

Lelusz [21] found that the compressive strength of cement composites decreased with the increase of the CNT dosage. This research used different dosage rates of MWCNTs, ranging from 0.00% to 0.12% by weight of cement. A suitable mix proportion was proposed by Manzur et al. [22] in a study aiming to produce surface-treated, MWCNT-reinforced cement composites based on their flexural and compressive strengths. It is, therefore, evident that carbon nanotube reinforcement could result in robust and improved cementitious composites.

Campillo et al. [23] experimented on a cementitious compound containing carbon nanotubes under compression and found that SWCNTs ( single walled nanotubes) and MWCNTs increased the compression resistance in relation to that of pure cement when at 6% and 30%, respectively. Lai and Bassem [24] conducted experiments to investigate the effectiveness of the carbon nanotubes, uniformly dispersed and randomly oriented, added for the reinforcement of cementitious composites. 

Results of the work by References [8,25,26] proved that the nanomodification of cementitious material could lead to significant improvement of its mechanical properties, compactness, and durability. Several researchers have been investigating the effects of different concentrations of CNTs on the strength of cement-based materials but no consensus has yet been reached [26].

This research focused on the investigation of the optimal incorporation of carbon nanotubes in cementitious composites. The objective was to optimize the cementitious composite design to evaluate the behavior of cement mortar produced by the addition of multi-walled carbon nanotubes (MWCNTs) in different concentrations by comparing their physical and mechanical properties with the corresponding properties of the nanotube-free material. It is important to highlight that the majority of published studies evaluating the ideal content of MWCNTs [20,21,22,26] randomly selected the number of mixtures used in the experimental procedures. Alternatively, this work uses the design of experiments (DoE) methodology to determine the number of mixtures. The aim was to obtain more accurate results based on an experimental design including destructive and non-destructive tests and SEM observations. 

## 2. Experimental Design

The experimental design was developed in the laboratories of the Universitat Politecnica de Catalunya (UPC), Barcelona, Spain, using destructive and non-destructive testing with mortar samples with and without carbon nanotubes.

The capacity to bear compression and tension forces were important indicators for the evaluation of the mechanical properties of the cementitious materials. Although the performance of these materials depended on several other factors, it provided a reliable indication of product quality. Low strength indicated that the mortar or concrete had problems in its structure that may have derived from the use of unsuitable material and may lead to the development of internal defects stemming from the lack of compaction or absence of proper hydration (cure procedure) [27].

### 2.1. Material Characteristics

This study was primarily oriented to obtain comparative results between cement mortar mixtures with and without carbon. Therefore, the following commercial materials typical of the construction industry were used in this study: Portland cement CEM I 52.5 R UNC-EM 197-1 manufactured by Cimento Molins Industrial S.A., Barcelona, Spain;Sand from Arids Catalunya S.A., Barcelona, Spain;Potable water from the Water and Sanitation Company of Barcelona, Barcelona, Spain;Superplasticiser (SP) polycarboxylate ADVA Flow 401 produced by GRACE Construction Products according to European Standard EN 934-2.28 [28];Carbon nanotubes produced and supplied by the Nanomaterials Laboratory of Physics, Department of Universidade Federal de Minas Gerais (UFMG), Brazil. The nanotubes were produced by chemical deposition in vapor phase and identified by MWCNT HP2627. They have the following characteristics: type—multi-walled carbon nanotubes (MWCNTs); weight—60 g, purity >93%; other carbon structures <2%; contaminants <5% of catalyst powder type MgOCo-Fe; and external diameter dimensions, 99% of the CNTs, between 5 and 60 nm and an estimated length between 5 and 30 µm.

### 2.2. Cement Mortar Compositions

To analyze the benefits of CNT incorporation, cement mortars were produced with a *w*/*c* of 0.5 with the carbon nanotube addition of 0.00%, 0.20%, 0.40% and 0.60%, by cement weight, and a constant sand/cement ratio of 3:1, according to BS EN 196-1 [29]. Table 1 presents the mixture compositions used to evaluate the behavior of cement mortars with different concentrations of CNTs.

### 2.3. Cement Mortar Production and Specimen Preparation

The cylindrical (44 mm × 80 mm) and prismatic (40 mm × 40 mm × 160 mm) specimens were casted according to European standard EN 196-1 [29], which recommend the standard composition for mortar production to be as follows: 1:3 cement/sand ratio and 1:0.5 cement/water ratio. The amount of superplasticizer (SP) and carbon nanotubes obtained was with respect to the cement mass. 

The specimens were produced in batches of six due to the small capacity of the mixer and according to the following procedure: (1)Cement and sand were weighed on a precision balance (Gram brand, model ST-4000, with a maximum capacity of 4000 g and accuracy of 0.1 g);(2)Sand and cement were mixed manually until acquiring a homogeneous appearance;(3)Water and SP were weighed on the same balance. SP and water were then mixed manually in a plastic container for about 5 min;(4)Nanotubes were weighed on the same balance. CNTs were then added to the water and SP mixture [Sikora 2018] and mixed by hand with a glass rod for 5 min;(5)Dispersion of the nanotubes and homogenization of the CNTs occurred and they were submitted to a physical and chemical procedure involving sonication, to deagglomerate the CNT bundles, for 60 min (see Figure 1). For this purpose, we used “ultrasonic P2000 clinging qteck Gmbh (Gemarny)” equipment;(6)Cement and sand (previously mixed) were poured into the mixer containing the mixture of the water, SP, and CNT after 15 min of mixing.

After completely mixing all the components, the cement mortar compaction was done in two layers using a manual vibrating platform. The specimens were kept for 24 h in a chamber at a temperature of 21.4 °C and 99% relative humidity. After 24 h, the specimens were demolded and returned to the humid chamber until the test date at 3, 7 or 28 days. 

### 2.4. Mechanical Properties

The compressive strength and splitting tensile strength were performed at 3, 7, or 28 days, according to standards of UNE-EN 196-1 [29]. These were adopted for this test and a hydraulic-digital machine, INCOTECNIC model PA/MPC-2 (Seville, Spain), with a 20 ton capacity was used, as shown in Figure 2. For each mixture and age, three samples (replication factor) were used, resulting in a total of 72 samples. More tests were conducted for the mixtures CN0 and CN4 to obtain the flexural strength (three-point bending test). This test was performed in prismatic specimens with the dimensions 40 mm × 40 mm × 160 mm at 28 days.

### 2.5. Physico-Chemical Performance and Non-Destructive Tests

To obtain the physicochemical properties, porosity tests were performed (see Figure 3). The pore structures of cement mortars incorporating variable nanotube concentrations were analyzed by vacuum porosimetry. Additionally, microscopic observations (SEM) were made to evaluate the microstructure characteristics, the composition, crystallographic and morphology. A Hitachi model S-4100 scanning electron microscope (Tokyo, Japan) was used. Samples for this test were obtained at 28 days. Furthermore, non-destructive tests such as ultrasonic pulse velocity (55 kHz) were done according to EN 14579 [30], to obtain the dynamic elastic modulus and the density of the cement mortar mixtures.

## 3. Results and Discussion

Aiming to obtain accurate results, this study employed the design of experiments (DoE) methodology [31] using the MINITAB program (Minitab 17), as can be seen in Table 2. 

### 3.1. Compressive Strength

The compressive strength of cement mortar samples was investigated at 3, 7 and 28 days. Table 3 presents the maximum and mean compressive strengths. The composite containing 0.40% nanotubes showed the best performance at all ages with a substantial increase in compressive strength compared to the reference sample CN0. The highest gain, about 42%, was reached at 7 days. The incorporation of carbon nanotubes in mixtures CN2 and CN6 did not increase the compressive strength. On the contrary, CN6 presented a decrease of 8.29% in compressive strength at 28 days. 

The fact that the CN4 performed better may indicate that there is an “optimal” range for the incorporation of nanotubes in cement composites. According to this study, the limit should be 0.40 wt % of Portland cement. We noticed that outside of that range there was no significant gain. On the contrary, there was a reduction of compressive strength. Some authors [32] have studied additions of CNTs between 0.5% and 1.0% mass of cement and found a decrease in compressive strength by increasing the content of CNTs to 1.0 wt %. Conversely, authors studying additions under 0.1% of CNTs, by mass of cement, have found optimum additions almost doubling the compressive strength of the composite [33]. In another study, an optimum dosage rate of 0.3% was utilized [34].

The noted increase of compressive strength was also related to the dispersion of carbon nanotubes in the mixtures [11]. When the dispersion was performed well, the carbon nanotubes were homogenously diluted in the cement paste by making interconnections with the hydrated calcium silicate and the grains of the mixture, with no agglomeration. This resulted in a denser matrix, which contributed to obtaining a new tougher material; similar results were presented by Reales and Filho [35].

### 3.2. Compressive Stress Versus Strain

The ductility of the cement mortar was investigated via the stress–strain curve. Figure 4a–d show the behavior of different samples and mixtures at the curing age of 3 days. In Figure 4a we observed that the sample set as CN0R6 showed irregular behavior, reaching a limit fc below expectations. All samples containing 0.40% carbon nanotubes achieved higher peak values of fc, at 3 days when compared to other blends of the same age, as seen in Figure 5c. Single peak behavior can be noted in the stress–strain diagrams of all specimens, which influences the strain capacity. Moreover, under compressive stress, the incorporation of CNTs did not lead to ductile behavior. However, Reales and Toledo Filho [35] reported that the ductility of the cementitious composite should be sensitive to the length of the CNTs. Therefore, the maximum strain was modified by the aspect ratio (diameter/length) of the CNTs.

Figure 4a shows the behavior of different samples and blends, at 28 days. Figure 4g there are no k, please modify shows that the largest peak values of fc, at 28 days of curing, corresponded to the blend with 0.40% carbon nanotubes. Figure 4e, shows that sample 1 of the mixture without carbon nanotubes (CN0) also presented, some preparation problems.

The compressive strength variation of each mixture at 3, 7 and 28 days is presented in Figure 5. The curves indicate the average fc values recorded, with a rupture of the three samples used in each trial. Figure 5c shows that the compressive strength of the mixture with the addition of 0.40% of carbon nanotubes significantly increased at all time points compared with the reference mixture (CN0), reaching the highest fc value at 28 days.

### 3.3. Tensile Stress 

Table 4 summarizes the results of splitting tensile strength. We observed that the mixture containing 0.40% nanotubes had the best performance at all ages with a substantial increase in tensile strength, compared to reference sample CN0. The tensile strength reached its peak, an increase of approximately 31%, at 28 days.

It is noteworthy that all samples with carbon nanotubes enhanced the tensile strength at 7 and 28 days. The CN2 sample reached a gain of 12.85% at 28 days in relation to the reference sample CN0. This result corroborates the effectiveness of the nanotubes for improving the tensile strength by acting in a significant way on the weak point of concrete, which is its tensile strength. Similar behavior has been reported in previous studies [36].

The fact that the sample CN4 performed better may indicate that there is a range considered “optimal” for the incorporation of nanotubes in a cement matrix.

We noted that the average value of tensile strength was around 2.93 MPa at 3 days of curing, increasing to 3.15 MPa at 7 days and reaching 3.56 MPa at 28 days. From the point of view of the variable, i.e., percentage of nanotubes, the average tensile strength was 3.0 MPa for CN0, increasing to 3.17 MPa for CN2, peaking at 3.71 MPa for CN4, and reducing to 3.0 MPa for CN6.

### 3.4. Flexural Load–Deflexion Relationship 

The tests for obtaining the flexural curves of the mixtures were performed on prismatic samples with the dimensions 40 mm × 40 mm × 160 mm. The test results are shown in Table 5.

As can be seen in Table 5, the fracture point of CN4 was at 2.99 kN, 15% higher than the rupture load supported by the reference mixture nanotubes (2.61 kN). This result corroborates an important characteristic of the carbon nanotubes, their high resistance to bending, resulting in a significant improvement in the well-known weakness of concrete. Additionally, we observed that from the beginning of the load applied to the point of rupture, the CN4 sample withstood a deformation of 0.90 mm, whereas with CN0 the deformation was 0.81 mm, indicating that the incorporation of nanotubes increased the composite ductility, i.e., greater deformation capacity, and could therefore withstand more loading. Li et al. [36] performed the flexural test in prismatic samples (40 mm × 40 mm × 160 mm), achieving a 25% increase in the flexural strength of mortars with 0.50% carbon nanotubes. Additionally, some studies [32,37] of 0.5% and 1% CNT reported that the increase in the flexural strength depended on both the filling of the pores of hydration and the tubes’ bridging effect, providing reinforcement to the matrix.

### 3.5. Non-Destructive Test (DME—Dynamic Modulus of Elasticity)

According to the results obtained, the CN4 and CN6 samples showed higher DME values than the CN2 and CN0 samples at all ages. The CN4 sample had the most expressive increase of DME in relation to the reference sample, reaching the peak value at 28 days and the highest gain, 25.53%, at 7 days.

The addition of 0.2% nanotubes to the concrete did not cause a significant change. On the contrary, the DME value was reduced at 28 days. Therefore, the sample containing 0.40% nanotubes was the one that presented the best overall performance at all the ages analyzed. The fact that this trait had the best performance may show that there is an “optimal” range for the addition of nanotubes in cement matrices. We noted that, outside this range, there was no significant gain but, instead, a loss of resistance sometimes occurred. Metaxa et al. [38] showed that the MWCNT-reinforced nanocomposites presented a higher modulus of elasticity compared to the reference cement paste samples.

The contour plot of the elasticity modulus test, Figure 6 shows the CNT content versus the age variables, all points where DME has the same value. The gradient of the function is always perpendicular to the contour lines. At 7 days, the DME value for CNT content, between 0.2 and 0.3, lay in the region defined between 22 and 24 GPa. The darker green hue defines the maximum points of the experiment. In another study, Hawreen and Bogas [39] found that the incorporation of CNTs (0.05% and 0.5% by cement weight) led to an increase in the elasticity modulus of concrete, emphasizing the effect of CNTs on microcrack propagation and nanoporosity reduction.

### 3.6. Porosity, Density and SEM Observations

The porosity tests were performed for a different mix at 28 days. The results, which include the percentage of pores (porosity) and the apparent and relative densities of the samples, are shown in Table 6.

The apparent density considers the effect of the pores and, hence, the amount of water contained in the sample volume, while the relative density excludes the influence of the water on the measuring process and is, therefore, higher than the apparent density. A similar trend, reported by Reference [38], indicated that the density was hardly affected by the addition of CNTs, probably due to the lower volume of this material compared to the overall amount of cement and sand.

For a comparative view of the average values obtained from the porosity tests, Figure 7 and Figure 8 show, respectively, the variation of the percentage of pores and that of the densities based on the addition of different amounts of carbon nanotubes to the mortar samples. The values of the real density and apparent density for the four samples were very close. However, the CN4 sample presented the highest values for both, indicating a denser microstructure, probably due to the filling of the pores and the better interconnection between the grains of the mixture in the presence of the nanotubes.

In the evaluation of the porosity of the samples, the CN4 blend also showed considerably better performance, with a reduction of approximately 15% of pores compared to the reference mixture CN0. This result led us to understand that the composite containing 0.40 wt % carbon nanotubes presented a denser microstructure, not only by filling the pores but also by interconnecting the hydration and producing smaller diameter pores. We also noted that samples CN2 and CN6 did not demonstrate significant improvements to their microstructures. A similar result was presented by Dalla et al. [39], perhaps because the nanotube entanglement was more evident with a higher content of the nanomaterial.

The decrease in the porosity and the increase in the density of the mixture with the addition of nanotubes was a positive factor for the durability of the concrete structures because with such a microstructure the movement of aggressive and deleterious agents within the concrete becomes more difficult [37].

The SEM micrographs obtained for the samples of concrete without nanotubes (CN0), at 1000×, 10,000× magnification, are shown in Figure 9. The images obtained for the sample CN0 validate the theory reported in the literature [32,40] regarding the formation of individual hexagonal acicular crystal-like columns. These are formed due to the formation of the Ettringite (AFt phase) or agglomerates, which represent one of the products of cement hydration—Ettringite. In addition, it is possible to observe the formation of some plaques, which indicate the presence of calcium hydroxide.

It can be noted that the incorporation of nanotubes in the matrix altered its morphology. When comparing the images of CN2 with CN0 at 10,000× times magnification, Figure 10, we observed that sample CN2 presented smaller acicular crystals than those formed in the sample without nanotubes. The presence of carbon nanotubes in the sample CN2 can also be observed at magnifications of 20,000× and 100,000×.

In the CN4 micrographs at magnifications of 5000×, 10,000×, 50,000× and 70,000×, shown in Figure 11, a higher concentration of nanotubes could be identified interlacing the hydrated compounds of the cement. Another important difference between the samples was the more homogeneous hydration in CN4, considering that the CN4 trace was formed with several acicular crystals, whereas in the sample without nanotubes these crystals were at localized points. Some researches [41,42] found similar aspects regarding the nucleating effect of nanotubes on the hydration of Portland cement.

The CN6 sample (Figure 12) also showed better morphology than the reference sample CN0 because the micrographs showed that the Ettringite crystals were more dispersed in the matrix of the compound CN6. The presence of carbon nanotubes also resulted in better bonding of the cement hydration products, relative to the reference concrete.

## 4. Conclusions

The incorporation of small percentages of carbon nanotubes led to significant improvement of cementitious matrix characteristics. According to the results obtained in the present study, the main conclusions can be presented as follows:The addition of 0.40 wt % of cement of carbon nanotubes (CN4) resulted in cement mortar with the best performance compared to the reference mixture, achieving an increase of approximately 40% compressive strength, 30% tensile strength, 15% flexural strength and a denser microstructure.The composites with the addition of 0.20 and 0.60 wt % CNT did not present the same level of improvement as the 0.40 wt % CNT, compared to the reference (0.00 wt % cement).The stress–strain curves of all specimens showed single peak behavior, which indicated the limited strain capacity of both the reference and the CNT mixtures, indicating that under compressive stress this material was brittle.The porosity measurements were performed to obtain additional information about the microstructure of the new compounds, incorporating nanotubes and 0.40% CNT produced the lowest pore percentage compared to other cement mortars.Analyzed SEM images showed the densification of the CNT cementitious matrices. This should be attributed to the lower porosity and the bridging effect of the carbon nanotube. As a result, better mechanical performance was obtained at 0.40 wt % CNT.

It is important to highlight that further studies of the environmental impacts and toxicity of new cementitious products containing nanotubes are recommended, together with the definition of handling standards and personal protective equipment for professionals working with nanomaterials.

## Figures and Tables

**Figure 1 materials-12-01504-f001:**
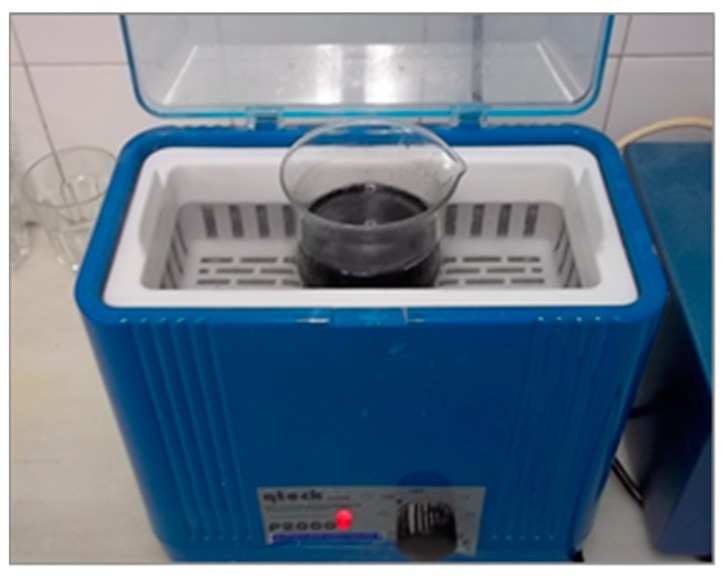
Dispersion of CNT by sonication. Please provide the higher resolution version

**Figure 2 materials-12-01504-f002:**
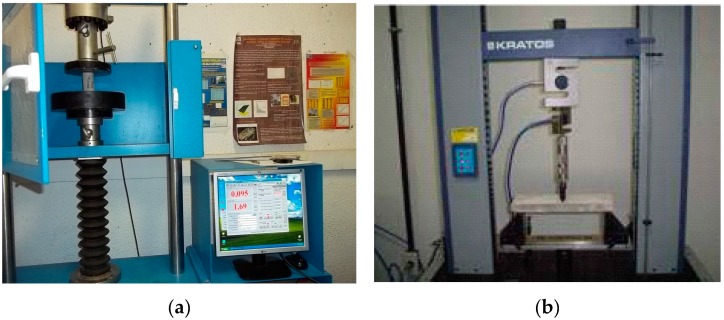
Universal testing machines: (**a**) compression and tensile strength tests and (**b**) flexural strength tests (source: Department of Material Science and Metallurgical Engineering, UB).

**Figure 3 materials-12-01504-f003:**
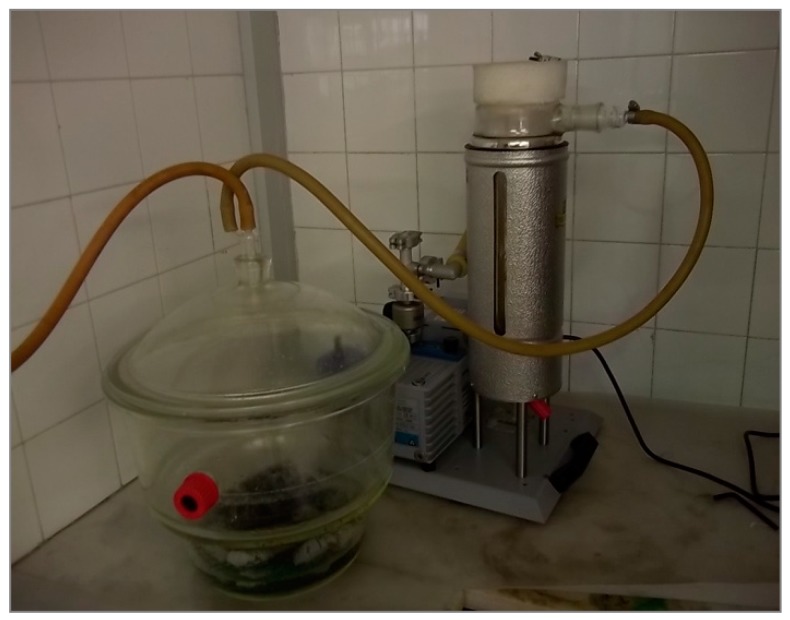
Porosity test equipment.

**Figure 4 materials-12-01504-f004:**
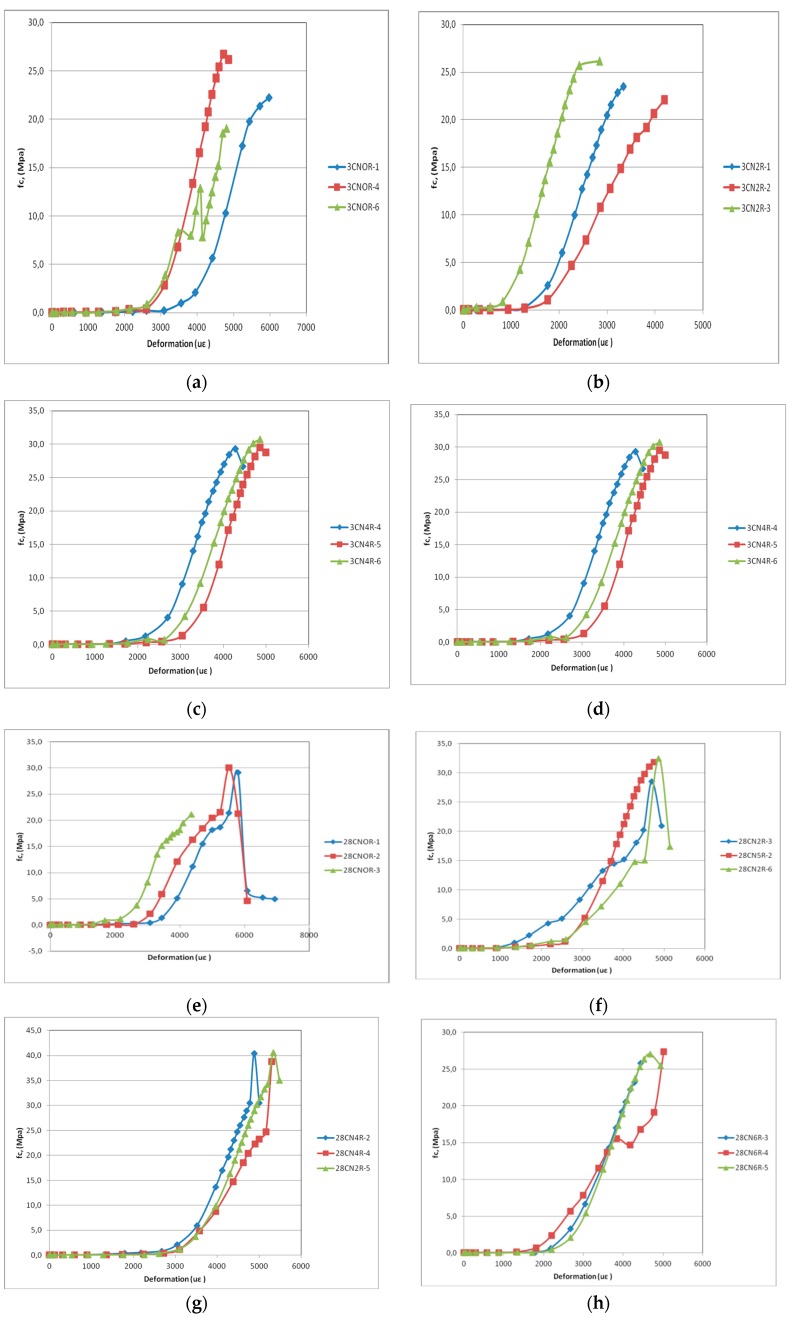
Stress–strain curves at 3 days: (**a**) 0%; (**b**) 0.2%; (**c**) 0.4%; (**d**) 0.6%; Stress–strain curves at 28 days: (**e**) 0%; (**f**) 0.2%; (**g**) 0.4%; (**h**) 0.6%.

**Figure 5 materials-12-01504-f005:**
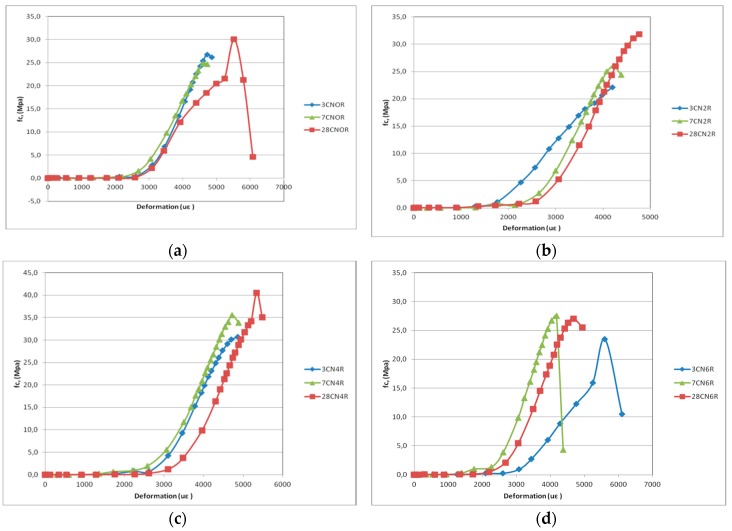
Compressive strength variation curves for each blend and age: (**a**) CN0; (**b**) CN2; (**c**) CN4 e; (**d**) CN6.

**Figure 6 materials-12-01504-f006:**
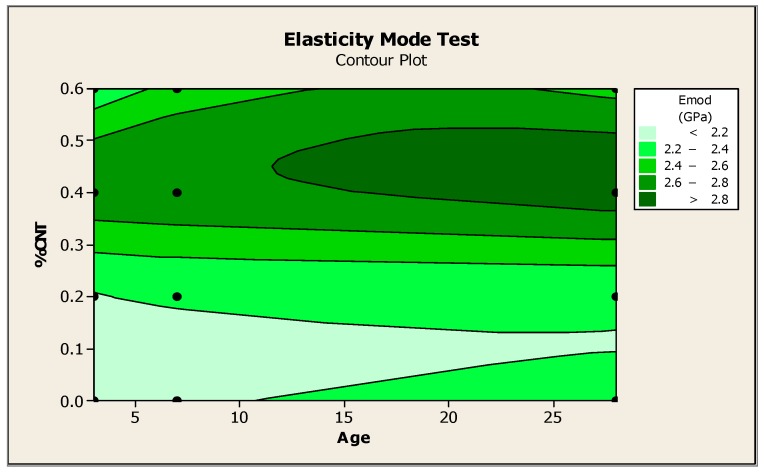
Level surface, in 2D, of the dynamic modulus of elasticity.

**Figure 7 materials-12-01504-f007:**
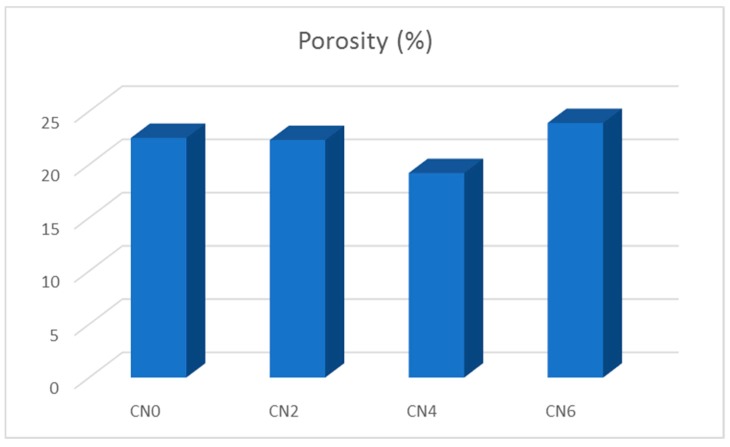
Percentage of pores in cementitious composite incorporating CNT.

**Figure 8 materials-12-01504-f008:**
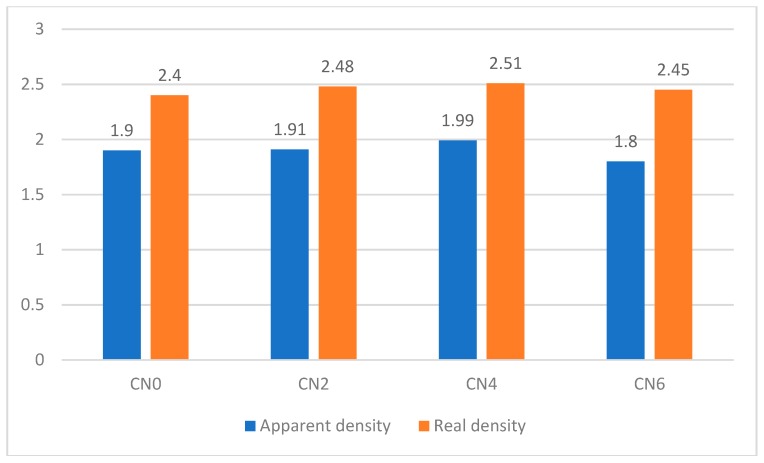
Apparent and real densities of cementitius composites incorporating CNT.

**Figure 9 materials-12-01504-f009:**
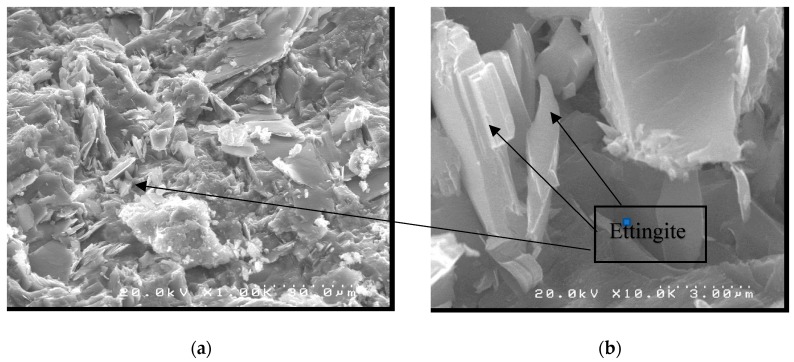
SEM micrographs of sample CN0, at (**a**) 1000× and (**b**) 10,000× magnification.

**Figure 10 materials-12-01504-f010:**
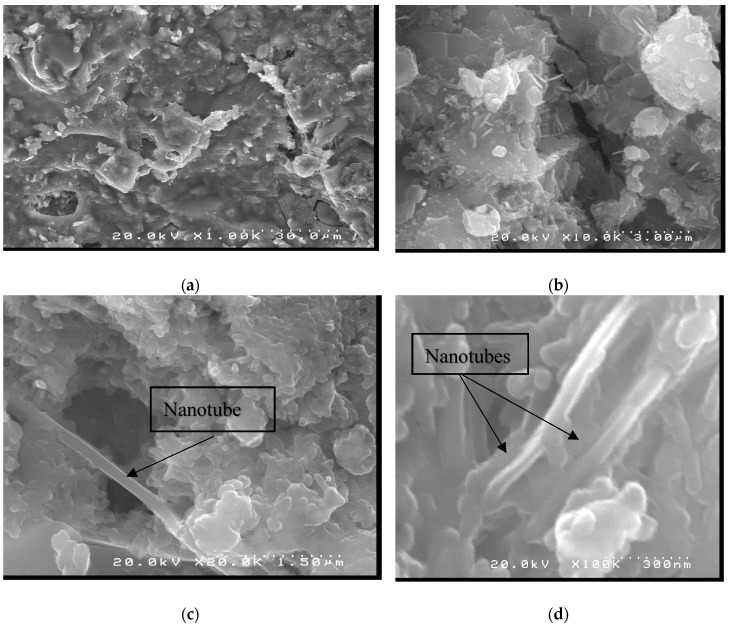
SEM micrographs CN2 at (**a**) 1000×, (**b**) 10,000×, (**c**) 20,000× and (**d**) 100,000× magnification.

**Figure 11 materials-12-01504-f011:**
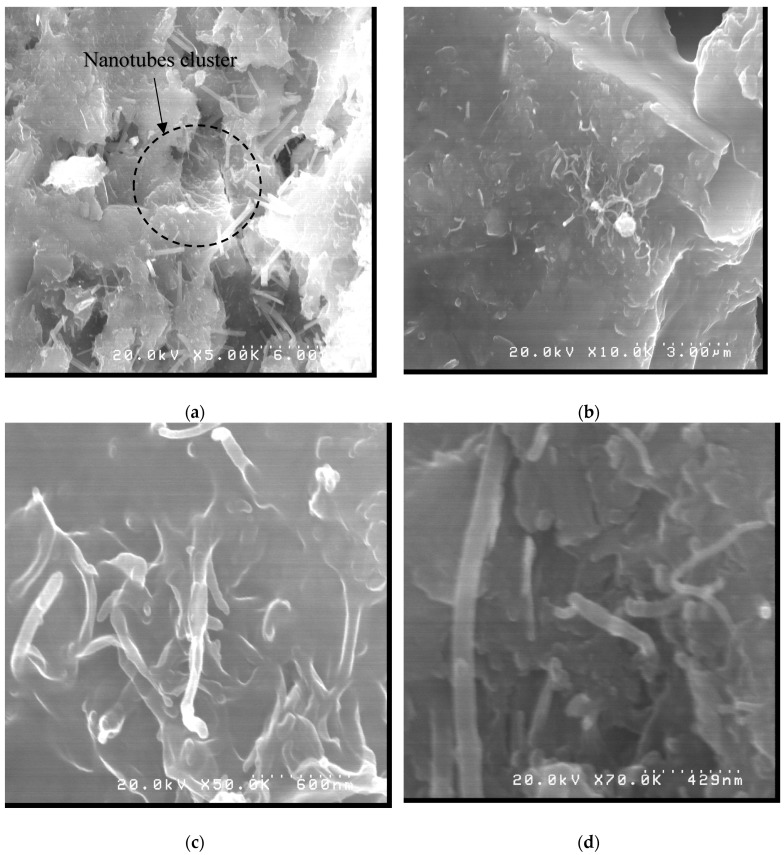
CN4 micrographs at (**a**) 1000×, (**b**) 10,000×, (**c**) 20,000× and (**d**) 100,000× magnification.

**Figure 12 materials-12-01504-f012:**
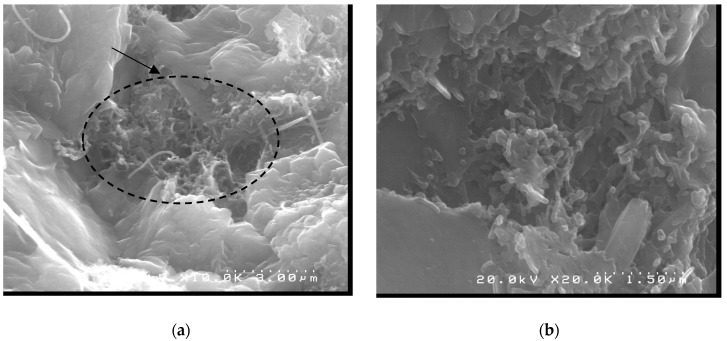
CN6 micrographs at (**a**) 1000×, (**b**) 10,000×, (**c**) 20,000× and (**d**) 100,000× magnification.

**Table 1 materials-12-01504-t001:** Cement mortar compositions (in grams).

Sand	Cement	Water	SP (1%)	Carbon Nanotube (CNT)
1200	400	200	4	0
1200	400	200	4	0.8
1200	400	200	4	1.6
1200	400	200	4	2.4

**Table 2 materials-12-01504-t002:** Design of experiment results.

MINITAB-Design Table								
StdOrder	RunOrder	PtType	Blocks	% CNT	Age	σ_c_ (MPa)	σ_tr_ (MPa)	DEM (GPa)
1	1	1	1	0.0	3	22.23	2.85	21.821
2	2	1	1	0.0	7	26.60	2.69	21.759
3	3	1	1	0.0	28	29.10	2.98	22.713
4	4	1	1	0.2	3	23.57	2.80	22.238
5	5	1	1	0.2	7	30.65	3.13	22.051
6	6	1	1	0.2	28	28.54	3.45	22.904
7	7	1	1	0.4	3	29.27	3.18	26.821
8	8	1	1	0.4	7	35.58	3.34	26.809
9	9	1	1	0.4	28	40.39	4.34	28.626
10	10	1	1	0.6	3	23.46	2.88	20.989
11	11	1	1	0.6	7	24.83	3.17	23.956
12	12	1	1	0.6	28	25.81	3.06	25.685
13	13	1	1	0.0	3	26.75	2.32	21.743
14	14	1	1	0.0	7	24.87	2.74	22.046
15	15	1	1	0.0	28	30.04	3.34	23.521
16	16	1	1	0.2	3	22.11	2.69	21.470
17	17	1	1	0.2	7	26.18	3.22	22.729
18	18	1	1	0.2	28	32.45	3.63	22.611
19	19	1	1	0.4	3	29.49	3.22	27.045
20	20	1	1	0.4	7	38.39	3.61	27.274
21	21	1	1	0.4	28	38.77	4.12	28.613
22	22	1	1	0.6	3	23.68	2.64	22.377
23	23	1	1	0.6	7	27.54	2.70	24.975
24	24	1	1	0.6	28	27.36	3.18	25.011
25	25	1	1	0.0	3	19.04	3.53	21.341
26	26	1	1	0.0	7	27.35	3.25	21.612
27	27	1	1	0.0	28	31.61	3.26	22.960
28	28	1	1	0.2	3	26.15	3.05	22.008
29	29	1	1	0.2	7	26.06	2.82	21.914
30	30	1	1	0.2	28	31.81	3.72	21.883
31	31	1	1	0.4	3	30.71	3.38	27.259
32	32	1	1	0.4	7	37.59	4.13	28.040
33	33	1	1	0.4	28	40.54	4.08	28.897
34	34	1	1	0.6	3	22.41	2.63	23.460
35	35	1	1	0.6	7	24.26	3.09	24.669
36	36	1	1	0.6	28	27.05	3.58	25.152

σ_c_—compressive strength; σ_tr_—tensile strength; and DEM—dynamic elastic modulus.

**Table 3 materials-12-01504-t003:** Maximum and mean values and gain of compressive strength.

Age	Compressive Strength (MPa) (Average)			
	CN0	CN2	CN4	CN6
3 days	22.67	23.94	29.82	23.18
7 days	26.27	27.63	37.19	25.54
28 days	30.25	30.93	39.90	26.74

**Table 4 materials-12-01504-t004:** Maximum and mean values and gain for the splitting tensile strength.

Age	Splitting Tensile Strength (MPa) (Average)			
	CN0	CN2	CN4	CN6
3 days	2.90	2.85	3.26	2.72
7 days	2.89	3.06	3.69	2.99
28 days	3.19	3.60	4.18	3.27

**Table 5 materials-12-01504-t005:** Flexural load–deformation relationship.

CN0		CN0		CN0		CN4		CN4		CN4	
Load (kN)	Def. (mm)	Load (kN)	Def. (mm)	Load (kN)	Def. (mm)	Load (kN)	Def. (mm)	Load (kN)	Def. (mm)	Load (kN)	Def. (mm)
0.0440	2.83	0.5405	3.06	1.7723	3.33	0.0314	1.66	0.1885	1.84	0.8107	2.22
0.0566	2.84	0.5719	3.07	1.8477	3.34	0.0314	1.66	0.2137	1.86	0.8233	2.24
0.0691	2.85	0.6159	3.08	1.9043	3.35	0.0377	1.66	0.2388	1.88	0.8610	2.26
0.0754	2.85	0.6662	3.10	1.9169	3.36	0.0440	1.67	0.2640	1.89	0.8862	2.29
0.0754	2.86	0.6976	3.11	1.9294	3.37	0.0503	1.67	0.3080	1.91	0.9050	2.31
0.0880	2.87	0.7542	3.13	1.9483	3.38	0.0503	1.67	0.3582	1.93	1.0307	2.34
0.1006	2.88	0.7982	3.14	1.9609	3.40	0.0628	1.68	0.3959	1.94	1.1375	2.36
0.1194	2.89	0.8673	3.16	1.9671	3.41	0.0628	1.69	0.4588	1.96	1.2758	2.38
0.1320	2.90	0.9239	3.17	1.9609	3.42	0.0691	1.69	0.4902	1.98	1.4141	2.40
0.1445	2.91	0.9679	3.18	1.9169	3.44	0.0691	1.70	0.5531	1.99	1.5209	2.42
0.1571	2.92	1.0307	3.20	1.8729	3.45	0.0754	1.71	0.6033	2.01	1.6780	2.44
0.1823	2.93	1.0873	3.21	1.7912	3.47	0.0754	1.72	0.6850	2.03	1.7912	2.45
0.2074	2.94	1.1627	3.22	1.6655	3.49	0.0817	1.72	0.7416	2.04	1.9357	2.47
0.2388	2.95	1.2381	3.24	1.6089	3.51	0.0817	1.74	0.8045	2.06	2.0740	2.48
0.2828	2.97	1.2884	3.25	1.5775	3.54	0.0943	1.75	0.8484	2.08	2.1682	2.49
0.3142	2.98	1.3701	3.26	1.6215	3.56	0.1006	1.76	0.8547	2.09	2.2814	2.50
0.3582	2.99	1.4329	3.27	2.0928	3.58	0.1131	1.77	0.8107	2.11	2.3631	2.51
0.3897	3.00	1.5083	3.29	2.3694	3.60	0.1257	1.78	0.8045	2.13	2.4699	2.52
0.4336	3.02	1.5838	3.30	2.4573	3.61	0.1383	1.80	0.7982	2.15	2.5768	2.53
0.4714	3.03	1.6466	3.31	2.5453	3.63	0.1445	1.81	0.7919	2.17	2.6396	2.54
0.5028	3.04	1.7157	3.32	2.6145	3.64	0.1697	1.83	0.7919	2.19	2.7276	2.55
										2.7842	2.55
										2.9959	2.56

**Table 6 materials-12-01504-t006:** Apparent density, density, and porosity.

Samples	Measurements			Results		
	Wet Weight (g)	Saturated Weight (g)	Dry Weight (g)	Apparent Density (g/cm^3^)s	Density (g/cm^3^)	Porosity (%)
CN0-S1	63.06	132.76	118.20	1.87	2.44	23.10
CN0-S2	65.34	140.49	125.98	1.93	2.48	22.20
Average	64.20	136.63	122.09	1.90	2.46	22.65
CN2-S1	55.96	120.02	107.53	1.92	2.47	22.30
CN2-S2	85.73	185.44	165.96	1.94	2.51	22.70
Average	70.85	152.73	136.75	1.93	2.49	22.50
CN4-S1	31.88	71.93	65.94	2.07	2.55	18.80
CN4-S2	60.12	132.41	120.52	2.00	2.50	19.80
Average	46.00	102.17	93.23	2.04	2.52	19.30
CN6-S1	39.64	83.42	74.15	1.87	2.44	23.40
CN6-S1	60.53	127.71	113.02	1.87	2.47	24.30

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
