# Peer review of "Evaluation of Carbon Nanotube Incorporation in Cementitious Composite Materials"

_materials, 2019, doi:10.3390/ma12091504_

Reviewer 1 Report

Please find attached a PDF with my comments and suggestions for authors

Author Response

1.      The state‐of‐art review performed in the introduction section is fine, although I suggest to clarify the aim and the novelty of the study improving the text of the paragraph in lines 97‐102.

Response 1: Our paper has been updated and includes a new paragraph in lines 122 - 131

2.      The materials and methods section is generally adequate. However, I find the subsection 2.2 a bit confusing. In order to make them more understandable, I suggest to divide it in several subsections, explaining in each one the tests performed one by one. This would clarify the experimental setup of the research included in the paper.

Response 2: We welcome the idea to divide in subsection and section 2 has been re-written and include subsection 2.1 , 2.2, 2.3, 2.4 and 2.5

3.      Regarding the results and discussion section, I think that it could be improved. I suggest to discuss the results with more depth and detail, because the majority of abovementioned section is used for describing the results, and the discussion of each one is relatively short. I also suggest to include more references in the discussion of results.

Response 3: The literature review has been improved and further discussion and comments are included in the revised manuscript.

4.      The conclusion section is very long, so I suggest to summarize it with the most relevant findings of the research included in the manuscript. In addition to this, the most important findings can be highlighted using bullet points or numbers, because it makes the conclusions clearer.

Response 4: The conclusion is improved and re-written to highlight the main findings. Sentences have been re-arranged accordingly in the revised manuscript to improve the readability

5.      Regarding the references, please include more references for supporting the discussion of the results.

Response 5 : We welcome your suggestion and new referefences were included in the revised manuscript.

Reviewer 2 Report

Dear Editor,

Thanks for inviting me to review Manuscript ID materials-475942 for your journal, please find my comments as follows:

This study is about incorporating MWCNTs in cementitious composites. Such a topic has been extensively investigated by researchers for the last few decades since its discovery on the early 1990s. The authors did not present anything new about introducing MWCNTs to the cementitious matrix. All the results are already well-known for the scientific community. The novelty/originality is missing in this work.

Author Response

1.      This study is about incorporating MWCNTs in cementitious composites. Such a topic has been extensively investigated by researchers for the last few decades since its discovery on the early 1990s. The authors did not present anything new about introducing MWCNTs to the cementitious matrix. All the results are already well-known for the scientific community. The novelty/originality is missing in this work.

Response 1. Currently, review papers state that even though direct benefits of CNT’s in composites have been identified, the mechanisms by which they interact with the hydration reaction to modify properties of the cement matrix are not yet fully understood. Additionally, considering the complexity of nanocomposites performance and the variety of mix designs and components materials, such as type of cement, admixture, cementitious materials (silica fume, fly ash, blast furnace slag) presented by previous researchers; several of the aspects discussed in the literature are still matter of open discussion, and require additional work. For example, CNT dispersion, stability, mixing procedure, hydration reaction, mechanical properties, durability and piezo-resistivity are some of them. Considering the fact, the construction is one of the most conservative industry to the adoption of new materials, and even more to incorporate cement-base nanomaterials, this paper contributes to the current discussion about the optimal concentration CNTs in cementitious matrix, based on experimental program established according to the Design of Experiment (DoE) method and analysing of both macro and micro properties of the cement mortar. Moreover, several researchers have been investigating the influence of different concentrations of CNTs on the strength of cement-based materials, but no consensus yet has been reached.

Reviewer 3 Report

The manuscript presents an experimental work aimed at properties the mortar containing CNT. There are following questions:

1.    Abstract and conclusions need to be rewritten to report about the main and new findings obtained in this paper briefly

2.    Please describe CNT processing and related physical properties.

3.    The results of the study should be further analyzed and mechanism discussion, and compared with other studies.

Author Response

1.    Abstract and conclusions need to be rewritten to report about the main and new findings obtained in this paper briefly

Response 1: We welcome your suggestion. Our paper has been updated and includes a new abstract and a new conclusion.

2.    Please describe CNT processing and related physical properties.

Response 2: A new paragraph has been written to describe CNT and its properties in lines 50 – 59 .

3.    The results of the study should be further analyzed and mechanism discussion and compared with other studies.

 Response 3: The literature review has been improved and further discussion and comments are included in the revised manuscript.

Round  2

Reviewer 1 Report

My comments and suggestions have been addressed, so I think that the manuscript can be accepted for publication.

Author Response

Thank you, we have appreciated all valuable comments and we are checking the English part for typos, style and correctness.

Reviewer 2 Report

The authors improved the manuscript and it is ready for publishing.

Author Response

(The authors gave the same response as above.)
